# Dealing with Cross-Sectoral Uncertainty: A Case Study on Governing Uncertainty for Infrastructures in Transition

**Oddrun P. Røsok \*, Mark L. C. de Bruijne and Wijnand W. Veeneman**

Department of Multi-Actor Systems, Delft University of Technology, 2628 BX Delft, The Netherlands
\* Correspondence: o.p.rosok@tudelft.nl

**Abstract:** The interdependencies between infrastructures are growing. Engineering decision making that earlier was largely confined to a specific sector now requires more and more understanding of how systems interact: a system-of-systems perspective. The article analyzes the effect of that added complexity in a single case study in de Zuid-As, Amsterdam, in the Netherlands, and relates the findings to the literature on engineering decision making and project management in complex projects. The article concludes that cross-sectoral engineering decision making has an additional level of complexity that requires governance of uncertainty. Despite this challenge being a well-known challenge among infrastructure operators, it is still not recognized for its importance, and it seems to be a neglected element in collaboration. Key is an open approach in the early stages that goes beyond classic cooperative decision making in engineering and project management environments.

**Keywords:** cross-sectoral; interdependency; infrastructures; uncertainty; governance; infrastructure projects

## 1. Introduction

Infrastructure projects are characterized by high complexity [1], which is understood here as the fact that many stakeholders are involved, all focusing on different elements of the project [2], understanding different parts and the whole of the project, as well as bringing different interests to the table when they make decisions. A great deal of literature is looking at the effective cooperation of these stakeholders to deliver a working asset. The literature covers different levels of stakeholder cooperation, dealing with the tensions that occur for example between the societal environment of the infrastructure or the infrastructure itself [3], between the broader infrastructure owner and the project team [4], the client and the contactor [5,6], or between subsystem contractors [7]. There are many other distinctions between perspectives, knowledge, and interests of stakeholders in which tensions through fragmentation can manifest themselves. As a side note, this research looks at how complexity manifests itself in the involved stakeholders rather than as a characteristic of the technical asset (see also [1]), because managing that complexity happens between those stakeholders with the technical asset being an outcome rather than a conditioning factor.

In cross-sectoral projects [8], this fragmentation is arguably larger than in intra-sectoral projects for several reasons. First, cross-sectoral projects are more unique. Similar intra-sectoral projects occur more often, with more (institutional) learning helping stakeholders overcome the limitations of their perspective, knowledge, and interest while working together on the construction of tracks or roads, cables, or pipes. Second, consequently, perspectives and knowledge on the whole-system level are likely to be more diverse in cross-sectoral projects, with different views from different sectors. Third, as stakeholders collaborate less often in cross-sectoral projects, the push for self-interests could be

more prevalent, as the long-term balancing of interests is less likely. So, the collaboration between large infrastructure organizations across sectors in joint projects possibly has an even greater level of complexity than normal infrastructure projects, with undiscovered issues of why it is difficult to work together.

This is a relevant issue, as the dependencies between infrastructures have been growing over time [9]. In the urban context, the shared spatial occupation is identified as a dependency. Growing urbanization incurs higher pressure on land use and on infrastructures. In addition, with the transition to electricity as a main energy source for mobility, this adds a functional dependency. Moreover, with growing reliance on digitalization, many infrastructures rely more and more on data networks. These interfaces are often recognized by pointing toward standardized couplings (such as 5G data, CCS car charging, ERTMS railway control, etc.), but the dependencies go much deeper: capacity levels need to align, provision needs to be coordinated, and failures propagate. At the same time, many critical infrastructure sectors are going through programs of refurbishment and transitions related to energy, circularity, and more. Infrastructure is becoming an extensive system-of-systems with a lot of dynamic complexity [10] to boot.

As sharing space and functional dependencies increasingly becomes a normal state of affairs in urbanized areas, solving conflicts around use of space pushes projects to solve these issues together. This article takes a closer look at the challenge of communication and coordination between infrastructure organizations while working on a joint project. The main angle is on how infrastructure operators deal with uncertainties when working on a joint project, and during the investigation, some key challenges regarding the collaboration were uncovered.

This article is a first inductive step into understanding the additional complexity that might exist in cross-sectoral projects. The article adopts a single case study approach and compares literature on complex project management to the reality of a cross-sectoral project. The article studies the integrated cable and pipeline tunnel (ILT) in the Zuid-As area in Amsterdam, in the Netherlands, to understand how the stakeholders carried out the communication and coordination between each other when uncertainties occurred. The ILT case was specifically selected because firstly, it had many infrastructure operators participating (e.g., drinking water, electricity, gas, telecom, etc.). Secondly, it was at that time highly innovative and the first of its kind in the Netherlands. Thirdly, it is assumed that there will be more projects of the same nature as the ILT in the future, where many infrastructure operators are involved in the same project and dependent on each other.
- Research question: How are cross-sectoral infrastructure projects governing uncertainty when coping with the added complexity of interdependent systems?

The goal of the in-depth case study of the ILT was to obtain an understanding of what happened during the realization of the project from when it was initiated back in 2003 until the summer of 2022, regarding the governance of uncertainties. To obtain such knowledge, a series of interviews were performed. The contribution of this article is that it shows the added complexity of large joint infrastructure projects. The literature says much about the challenges but lacks the empirical data to show the nuances and complexity of the reality of such projects.

The paper is structured as follows. Section 2 is a literature review on the challenges of dealing with uncertainty in a joint infrastructure project. Section 3 explains the research design and case background as well as how data were collected and analyzed. Section 4 presents the results of the study. Section 5 discusses the results, and Section 6 shows the conclusions of the study.

## 2. Literature Review

After a first analysis of the literature as well as talks with key stakeholders, four areas of literature were chosen in which the challenges of dealing with uncertainty manifest themselves the most when it concerns a project of cooperating (infrastructure) organizations. Those aspects are: shared goals, shared information, shared procedures, and a

shared perspective of the overall system. This formed the basis for an in-depth literature study.

The scientific literature was chosen through searching the Web of Science and Google Scholar with the use of search strings with keywords relevant to the fields of literature, which are as follows:

- Goal setting: collaboration and infrastr* and projects and (goals and/or objectives) and (setting and, or common and/or joint);
- Information sharing: collaboration and infrastr* and projects and (information and/or data) and (sharing and/or coordination);
- Joint procedures: (collaboration or coordination) and infrastr* and projects and (joint and/or aligned) and (decision making and/or procedures);
- Systems thinking: (collaboration or coordination) and infrastr* and projects and (systems and/or thinking).

A selection of articles was made based on the articles with the highest numbers of citations. A minimum citation limit was set for each literature field in order for the literature to be considered. The chosen articles are a result of the subsequent scanning of abstracts. For each field, a total of six to seven sources were used. Additional searches were conducted in Google Scholar through the 'snowball' effect where a relevant finding of literature lead to more relevant literature to ensure a satisfactory number of sources for the literature study.

Perrow [11] identifies two dimensions which affect the capability of organizations to provide sustained reliable operation of technologies. One of them is interactive complexity (the classic perspective of interactions between elements in the complexity literature), and the other dimension is the amount of coupling (the dimension considers whether dependencies between system elements are loose or tight). Perrow [11] claims that from an organizational perspective, the level of coupling and decoupling between the elements of a system forms a crucial factor which in turn influences the speed with which a failure can propagate through a system, which would defeat the possibility to mitigate failures, thus increasing the propensity for cascading failure resulting in ('normal') failure. Understanding the full interactive complexity is near impossible in highly complex systems.

The subsequent four fields were selected from a wider range with a focus on their function to analyze empirically the project under study. There is a great deal of normative theory on the management of uncertainty in classic project management literature, such as for example risk management literature. Risk management literature focuses strongly on analytical tools [12]. The field largely ignores organizational processes and cognitive limitations [13]. With that, we position this research in line with that of [14] with a strong empirical focus, including organizational mechanisms which identifies implications that go beyond the quality of any (application of) an analytical tool for managing the complexity of cross-sectoral projects.

### 2.1. Goal Setting

Organizations are goal-directed [15] in [16] (p. 735), where the goal setting is about specifying one's interests, focus, and end-result. The same holds true in a collaboration where goals are set together. Goals can also be the "reasons why collaborations are initiated, what organization participants aspire to achieve, and the nature of the collaborative advantage sought" [17] (p. 733). Goal setting goes hand-in-hand with collaborations where multiple people or organizations work together toward a common goal [18]. This process is achieved by "sharing knowledge, learning, and building consensus. In collaboration processes, individuals or organizations create relationships" [19,20]. Coming together to figure out the shared end goal for everyone is a process that brings people and organizations closer during that time. It assumes a level of information sharing and trust in the other participants to stay true to their goals. One element in this process is that "Collaborative action will be difficult to accomplish, much less assess, if shared goals and

an operating rationale for taking action are not made explicit" [21] (p. 17). This means that the goals need to be explained as clear as possible.

### 2.2. Information Sharing

Collaboration and coordination between organizations assumes a degree of information sharing when working toward a common goal both internally in an organization but perhaps more importantly between organizations. Sharing information can lead to closer collaboration if the "[…] individuals or groups are recognized and rewarded for sharing their ideas and expertise with other members within the organization" [22] (p. 86). This means that people are more likely to share information if it is perceived as beneficial. At the same time, the opposite holds true: that sharing information or knowledge is seen as unnecessary if the benefits of sharing are not made explicit [22] (p. 86). Interestingly, it is also paradoxical to share information with other organizations. On one hand, an organization may want to share information, but on the other hand, the organization wants to protect information that can be advantageous to them [23] (p. 4). Nevertheless, the importance and benefits of sharing information is already well-established in the literature [24,25] in [26] (p. 39). Ref. [27] and [26] (p. 18) turn it on the head and say that opportunities in collaboration may be lost due to poorly managed information or knowledge. Sharing information or knowledge is difficult without a level of trust between people and organizations, and that trust is crucial for people to open up and start building partnerships [28] (p. 568).

### 2.3. Joint Procedures

The concept of joint procedures goes hand-in-hand with collaboration. Each organization in a collaboration has its own procedures and way of working. When multiple organizations come together who all have their own perspective, there is a potential for conflict. Joint procedures are about the way of working in a collaboration, so if the organizations have different ways of working and approaching a project, or they have to work in a new environment, then aligning all the different procedures becomes a challenge. This challenge is linked to information sharing [29,30]. The participants in a collaboration know little of each other's procedures and mindsets in the project, so aligning procedures assumes some communication. If the procedures take into consideration the "[…] needs of the group, implementation of the partnership is based on the participants, open and frequent communications occur to promote common understanding, and decisions are made through a participatory process" [31] (p. 392). This means that making an effort to understand the other participants' procedures, and collectively making procedures together to reach their common goal entails open communication between the organizations [32]: "[…] collaboration occurs when people from different organizations produce something together through joint effort, resources, and decision-making, and share ownership of the final product or service" [33] in [32] (p. 367).

### 2.4. Systems Thinking

The studies in the literature have a difficult time defining systems thinking [34]. There are many definitions, but the perspective that resonates the most with this research is that "[…] systems thinking examines relationships between the various parts of the system" [35] (p. 6). Systems thinking here is in other words about coordination and understanding and connecting the different contributions and perspectives in a project, of each participant in a collaboration. Each infrastructure organization is a system that brings with it its own way of thinking about its own system. Systems thinking points toward understanding the different systems as subsystems of a larger system: the collaboration [36]. "Boundaries must be set to distinguish what parts of the world are contained inside the system and what parts are considered the environment of the system" [35] (p. 6). At the same time, "Systems thinking accepts that social knowledge is provisional and context

dependent. It entails consideration not only of the desired outcomes but also of the complex web of inputs, processes, and outputs that lead to these outcomes" [37] (p. 724). This shows that systems thinking is flexible and has room to consider "[…] innovativeness, complexity and uncertainty…" [38] (p. 396). Large infrastructure projects are indeed innovative, complex, and filled with uncertainty, so it seems reasonable to use systems thinking since it "[…] offers the capacity to manage diverse settings and creates the discussion space amongst participants from diverse perspectives to listen and hear each other sharing their expertise" [39] (p. 1138).

### 2.5. Uncertainty

Lastly, a clarification on how this article defines uncertainty is helpful. Uncertainty is a well-known concept deferred from risk and has already been defined in the literature (e.g., [40–45]). There are many definitions, depending on one's perspective and context. The definition that this article will use as a starting point, is that uncertainty is "an event or situation which was not expected to happen, regardless of whether it could have been possible to consider in advance" [44] (p. 77): in other words, an 'unexpected event'. Both this phrase and 'uncertainty' will be used interchangeably to describe the same concept in the article.

## 3. Research Approach

### 3.1. Research Design and Case Background

This exploratory research makes use of historical empirical data and interviews with experts to analyze the complexity and challenges of cross-sector infrastructure project collaboration and specifically study how infrastructure operators dealt with uncertainty. The literature at this moment does not reflect the full complexity of the reality of joint infrastructure projects. To uncover such data, the research makes use of an abductive approach [46] and uses a grounded theory method to confront case observations with findings from the literature. "An understanding of the characteristics and consequences of case studies based on abduction [thus] requires an integrated approach, because the main difficulty of case studies is handling the interrelatedness of the various elements in the research work" [47] (p. 555): in other words, seeing the differences and similarities between the empirical data literature. This is in line with the argument that "[...] abduction reflects the process of creatively inferencing and double-checking these inferences with more data. As such, abduction fits in with the traditional grounded theory recommendation to move back and forth between data and theory iteratively" [46] (p. 168). However, combined with the abductive method, it lays a foundation for which one can construct theory [46] (p. 169).

This research makes use of an in-depth single case study approach: of a multi-actor infrastructure project. The project was presented to the researchers by the main representative of the infrastructure partner consortium which the authors were collaborating with. The case was a project that included many infrastructure operators and stakeholders who worked on a complex cross-sectoral infrastructure project in the heart of the business district Zuid-As of Amsterdam South in the Netherlands. The primary objective of acquiring data was to gain an understanding of the challenges of collaboration between infrastructure operators in a joint project. It is assumed that infrastructure operators will face more joint projects in the future, and this makes the selected case study interesting to study. At the time, the project was an innovative pilot project that challenged the infrastructure operator's ways of working and required collaborating. The added value of researching an innovative pilot project as a case study is the assumption from a research point of view that especially in these types of projects, sensitivity to uncertainties would influence the collaboration between infrastructure providers, which in turn would show in the governance of the project.

The single case study approach allowed for an in-depth study of how uncertainties were dealt with during the design and subsequent realization of the ILT project. This in

turn provided insights into the collaboration between the stakeholders. This is important because this article aims to uncover underlying 'hidden' professional, personal, and institutional forces which affect people's choices when coping with uncertainty. In other words, it means uncovering the forces which make it challenging to communicate and coordinate with each other regarding unexpected events.

Access to data and case documentation was a contributing factor to why a single in-depth case study was chosen. The chosen case study was finished in 2005, but some workers were still working in the same organizations and were available to share their experiences. Due to the time between finishing the project and this research, much of the case documentation was available and accessible.

### 3.2. Data Collection and Analysis

In addition to access to case documentation, empirical data from the case study were collected through nine semi-structured open-ended interviews with experts in the project, who were key personnel who had either actively participated in the project or were introduced at a later stage of the project, i.e., after the tunnel had been taken into use and operations started. The stakeholders involved in the ILT and their assets are listed in Table 1.

**Table 1.** Stakeholders and their assets.

| Infrastructure Operator | Infrastructure Asset(s) |
| --- | --- |
| Waternet | Drinking water, sewage |
| KPN | Telecom, phone, glass fiber (internet) |
| COLT | Telecom, phone, glass fiber (internet) |
| Continuon | Gas, electricity |
| NUON | City heating (water), city cooling (water) |
| Municipality of Amsterdam | Tunnel |

Interviews were conducted with five different managers from Alliander, the current energy infrastructure operator which at a later point merged the services of Continuon and NUON. These interviews provide a particularly rich perspective of the processes inside the gas and electricity side of operations, which researched in particular to obtain an in-depth understanding of the inner workings of infrastructure operators. Additional interviews were held with representatives from the municipality of Amsterdam, Waternet, as well as a short email correspondence with a manager from COLT Technology Services BV. Some of the interviewees were interviewed twice to obtain more insight into specific aspects, and some of the interviews were double, with two interviewees present at the same time. Lastly, there was a ninth interview conducted with another researcher who had also studied the ILT in a similar setting, providing context information. Due to the age of the project and limited access to people who worked on the project, the interviews ended up being overrepresented by Alliander. Nevertheless, all the major stakeholders, except for KPN and the brief correspondence with COLT, were represented in the interviews. The respondents provided project documentation, in addition to readily available documentation online. This documentation includes reflection reports, evaluations, financial analysis, presentations, risk analysis, transfer documentation, newsletters, and the user manual of the ILT.

The interviews were an important source of empirical information on the ILT as they gave insight into the experiences of the people who were deeply involved in the project in various roles, such as project management, asset management, risk management, and external relations management. Interviewees were found through the main contact between the researcher and the NGInfra Consortium. Some respondents mentioned other possible respondents who were involved in the ILT. This 'snowball-effect' method was

subsequently used, where an interviewee would give the contact information of others who they thought were relevant to talk to regarding the ILT project.

An interview protocol was made and used in every initial interview to remain consistent on the data collection. The focus of the main interview protocol was to uncover the interviewees experience in how their infrastructure organization had dealt with the governance of uncertainty internally and together with the other infrastructure operators involved in the project, if at all. The interview protocol also focused on potential challenges the interviewee saw, or could still see, regarding collaboration with other infrastructure operators. Even though the interviewees were potentially experiencing hindsight bias, they were still encouraged to share their reflections and thoughts on what they thought could be achieved differently in the project. Follow-up interviews had a new set of questions tailored which would go further in depth on the topics at hand, seeking additional information on said topics.

### 3.3. Analysis

The interviews were performed either face-to-face or through Microsoft Teams due to COVID restrictions. The interviews were recorded and transcribed using the edited transcription method as outlined in [48]. The data from the interviews were analyzed through sorting the responses (open coding) and subsequently putting them into clusters of themes (axial coding). This information was put in tables to connect the concepts of coordination to their manifestations in the ILT case (selective coding). Several manifestations were identified as well as their underlying mechanisms and overarching factors for what possibly caused the manifestations during the collaboration.

### 3.4. Empirical Case Context

The integrated cable and pipeline tunnel ('integrale leidingentunnel', or ILT) is located under the green belt of the Gustav Mahlerlaan between the 'Buitenveldertselaan' and the 'Beethovenstraat', in the Amsterdam South (Zuid) district in the municipality of Amsterdam [49]. The ILT was an innovative pilot project initiated by the municipality of Amsterdam and was among the first joint service tunnels to be developed in urban areas in the Netherlands. It is a double underground concrete tunnel that was finished in 2005. The ILT holds pipes and cables of many utilities such as gas, electricity, hot and cold water, sewage, rainwater, telecom, and Internet (glass fiber). It has since completion continuously evolved. Infrastructure operators install or uninstall infrastructures in the ILT regularly, as they are not allowed to install new infrastructures anywhere else but in the ILT if they want their infrastructure to run in this specific area.

At the moment of design and its completion, the ILT was considered an innovative pilot project. It was the first integrated utilities tunnel of its kind in the Netherlands, and there certainly were uncertainties to deal with both internally and between operators. From a research perspective, it was assumed that the innovative aspect included a new way of working or collaborating between infrastructure operators, and that especially given its unique and innovative character and its location in the center of the Zuid-As, uncertainties and the governance of uncertainty would be high on the agenda throughout the duration of the project. The ILT case was thus used to obtain insight into how governance of uncertainty takes place in a cross-infrastructural collaborative project setting.

### 3.5. The Integrated Cable and Pipeline Tunnel Case

The cable and pipework tunnel project (ILT) is located under the street of Gustav Mahlerlaan in the district of Amsterdam Zuid, just a minute walk from the Amsterdam Zuid station [49] (p. 6). The 'Zuid-As' district in Amsterdam is "[…] one of the most important office locations in the Netherlands", [50] and is an area with "[…] very high building density, high quality and little public space" [51] (p. 2). This means that this area is very busy and has a high density of utilities resulting in scarcity in space above and below

the ground. Consequently, conflicting claims for cables, pipelines, wires, and other utility infrastructures were very much anticipated. The maintenance, removal, and installment of infrastructures require that the infrastructure operators have access to said infrastructures. However, in an area such as Zuid-As, it was deemed not desirable by the municipality of Amsterdam to lay cables and pipes under the subsurface below street level, since the anticipated changes of the Zuid-As and subsequent digging would interfere with the ambition of the area to stay high-end and attractive as a business hub. Continuous work out in the open would affect transport mobility (i.e., traffic) and accessibility of the area for people [51] (p. 2) as well as affecting the atmosphere and appearance of an area that aims to generate high yields [51] (p. 4). Cables and pipework require space and are typically installed in the ground where the free spatial environment is now continuously decreasing. Some areas can also be less suitable for such installations due to greenery such as trees.

To deal with this infrastructure challenge, the municipality of Amsterdam came up with the idea of an underground infrastructure tunnel where infrastructures could be installed vertically. That way, the cables and pipelines are stacked in height rather than placed side by side, which ultimately reduces the use of horizontal space [51] (p. 4). This would free up space in the public space in the construction area. A finished tunnel such as this would also have negligible impact on the public space since all maintenance, removal, and installment of infrastructures would happen underground inside the tunnel. The concrete tunnel is roughly 500 m long, 6.5 m wide and 2.3 m high. It has two channels (north and south), and each channel has an open walkway. The cables, pipes and wires are installed on the walls, where some are resting on shelving systems. Rainwater drainage is installed underneath the tunnel, and the sewer is installed next to the tunnel on the outside. The tunnel can be accessed in the middle, where the technical room is located. The sewage pumping station is also located near the technical room and has a separate entrance [49] (p. 6).

## 4. Results

The following tables provide an overview of the main findings from the analysis of the data obtained during the case study of the ILT with a subsequent description of the content in the tables. Tables 2–5 present a condensed summary of the conceptual findings and the articles' analysis of these concepts. The findings were divided into four main themes during the coding of the interviews, and each theme represents a factor. The tables provide an overview of the connection between the factor, the literature field to which it connects, the type of mechanism that played a role, and how the conflict type identified in the literature manifested itself in the case.

**Table 2.** Time pressure and rushed decisions.

| Factor | Conflict Type Field of Literature | Mechanism | Manifestation of the Conflict Type |
|---|---|---|---|
| Time pressure and rushed decisions | Goal setting 'What do we want to achieve?' This is about preference - Differences in interests, incentives, values, understanding, and needs | Focus on output rather than uncertainty, design, and process 'Fixing' rather than 'integrated solutions' as a description of 'collaboration | Little influence on the project, and the boundaries of it Heavier focus on the finances than the assets Lack of joint planning |

The ILT was initiated by the municipality and presented to the stakeholders who were informed that if they wanted their infrastructures in the Zuid-As area they had to put their infrastructures in the tunnel, and they had to 'do it now'. In other words, the

'solution' was somewhat abruptly presented to the stakeholders, who had less time than preferred to figure out how to execute the project. Consequently, there was time pressure on all stakeholders. One of the stakeholders decided to 'sit on the fence' for a while, but when they finally decided to join up, the project had in the meantime moved forward. This meant that they had even less time to come up with their ideal solution to their assets. Due to the time pressure and innovativeness, some decisions were rushed and in hindsight less optimal.

This factor is linked to the literature of collaborative goal setting, that is about what the participants want to achieve in the project. The stakeholders had their own goals for what they wanted to achieve. In this collaborative setting, the involved infrastructure organizations did not really discuss how they could achieve their goals together. Instead, they set their own goals and worked in parallel.

Due to the stakeholders working in parallel rather than in actual collaboration, their focus was on the output, the asset(s), and finishing the project within the (tight) deadlines set by the municipality of Amsterdam. This way of working leaves little room to discuss uncertainty (what you do not know or are unsure about), joint design and join processes. When uncertainties showed up, as they always do, the mindsets of the stakeholders were focused on fixing the issue as it occurred instead of seeking integrated solutions together, by including the other stakeholders. It also became evident during the interviews that the infrastructure operator's understanding of uncertainty was 'not knowing how something would turn out'. This is different from 'an unforeseen event'. Once they understood uncertainty as an unforeseen event during the interview and were subsequently asked about their process of dealing with this understanding of uncertainty, the interviewees all had the same message: there is no established process in place to deal with uncertainty. One respondent asked rhetorically if it was possible to plan for something one cannot plan for. When the respondents were then asked about how they dealt with unforeseen events in the ILT project, their overall response was 'if there was a problem, we improvised'.

The way we could see the conflict arising in the ILT was that the participants from the beginning did not have much influence on the project and its boundaries. The municipality had already made the design of the tunnel and wanted it made that way, and the infrastructure operators had to try to fit into it if they wanted their infrastructure assets in this area at all. Bear in mind, the Zuid-As is a very lucrative area to have an infrastructure installed, meaning the infrastructures are highly valuable. During the realization of the ILT, the stakeholders kept focusing on the financial aspect of the tunnel even after the municipality had ensured them that the municipality would cover the extra costs beyond what they would pay for a 'standard' project. A 'standard' was defined as a project that the operator has accomplished many times before, working within a frame that is already familiar and predictable for each infrastructure operator. In other words, the stakeholders had agreed upon the project costing 'no more than what they normally would pay' in a regular project setting. Still, their focus was on finances, which in turn meant less time to focus on the functional solutions of their assets. Aside from the tunnel already being designed before the stakeholders were involved, after they had joined, they did not prioritize joint solutions which could benefit not only themselves but possibly also some of the other stakeholders.

**Table 3.** Trust.

| Factor | Conflict Type/Field of Literature | Mechanism | Manifestation (of the Conflict Type) |
|---|---|---|---|
| Trust | Information sharing 'What must we know?' This is about perspective | Trust issues as a hindrance that disallowed collaboration regarding the exploration of uncertainties | Lack of systemic understanding of each other's roles and needs |
| | Focus on own infrastructure | | Little collective effort to build trust |

| | Little insight in other organizations' 'inner workings' | |
|---|---|---|
| Paradigm differences of organizations | | Islanding |

Trust between organizations and people takes time. Within the ILT case, trust was thought of as 'I trust you to do what we have agreed upon'. The infrastructure operators did not easily share information with each other or give other stakeholders insight into their own organization. They played with their cards close to the chest.

The field of literature this is linked to is information sharing. Every stakeholder had their own perspective on their infrastructure and the ILT, but since they played with cards close to their chests, they did not easily share information about their plans or designs easily to the other stakeholders. This in turn makes it even harder to explore and talk about how to deal with uncertainties. In fact, they did not have a system in place to deal with uncertainty at all.

The mechanism was that trust, or in this case the lack of it, hindered collaboration between the stakeholders. This in combination with little insight into the 'inner workings' of the other stakeholders in the collaboration made it challenging to explore uncertainties.

There was a lack of 'systemic understanding' of the challenges of the project among representatives even within an organization, let alone across organizations. This means that people who worked with each other but came from different groups with different responsibilities in the project did not fully understand the position and perspective of other people's positions. To illustrate, a project manager (asset owner) thinks more of the company image and political position of the company, whereas the asset manager is more concerned about safety within the project. If, e.g., the project manager overstepped into someone else's 'domain', such as that of the asset manager, and thus included themselves where they were not supposed to, there would be friction. Communication was difficult and at times frustrating for the people involved because of differences in perspectives and overstepping of boundaries. The lack of systemic understanding also occurred between the stakeholders. The lack of trust in each other caused discomfort and unwillingness to open up and talk to each other. Little effort was made to build trust internally and externally between the organizations. As a side note, in the interviews and documentation, there was no talk of past situations causing the present lack of trust in each other. The 'lack of trust' is here meant to be understood as initial trust not being built among the stakeholders from the start. There was an ambition, where some CEOs would meet each other for a cup of coffee to build some trust, and this had an effect on the rest of the organization so that some felt more at ease. However, these efforts did not have much impact on the overall collaboration. The greatest impact came from a few individuals with the intention to share information and ideas on difficult topics, such as risks or even uncertainties. The lack of trust internally and externally caused something that can be described as 'islanding'. That is, individuals, groups of people, and organizations primarily focus on their own part of the job in the project and show little interest outside their 'island': that is their work domain and the task at hand. The concept of islanding could be seen both internally in organizations and between the stakeholders in the ILT. The intention in the project was to work together in collaboration, but instead, they worked alone in parallel.

**Table 4.** Prevalence of own procedures.

| Factor | Conflict Type/Field of Literature | Mechanism | Manifestation (of the Conflict Type) |
|---|---|---|---|
| Prevalence of own procedures | Joint procedures 'How do we decide?' This is about the way of working | Unwillingness to discuss other options/uncertainty | Unwillingness to adjust |
| | | | Lack of knowledge and understanding of the project |
| | Traditional infrastructures in an innovative environment | Disinterest or 'discomfort' of something new and unknown | Wishing to work as traditional as possible, but in a new environment |

The ILT was an innovative pilot project, and that assumed from a researcher's perspective that the infrastructure operators engaged in a different way of working than one normally would in standard infrastructure projects. The infrastructure operators, however, seek as much as possible to work in their traditional way(s). They wanted to execute the project in the way they were used to and had a prevalence of their own procedures. Throughout the project and across the various infrastructure operators, there was an unwillingness to change their traditional way of working as well as opinions.

This factor is linked to literature on joint procedures: how to create procedures together. There is something uncomfortable in trying something new and unknown, especially in an innovative pilot project that already is under time pressure and holds some prestige due to its location. The stakeholders wanted to work in the way they know and are comfortable with, but it does not mean that it is the best for the project outcome and collaboration. This 'discomfort' in stepping out of one's comfort zone also affected the discussion, or rather almost removed the option all together, of exploring other options of procedures and what procedure to follow in case of emerging uncertainties.

The type of conflict manifested itself through the unwillingness to adjust procedures beyond what the stakeholders were used to. As an example, to illustrate, one of the stakeholders in the ILT had a 'standard principle of obstinacy' toward the project. This means that the operator would disagree to the plan(s) from the beginning on principle. They did not want to adjust their own procedures to the join the project. This, in combination with a lack of knowledge and understanding of the ILT project, resulted in holding on to their traditional and 'comfortable' way of working.

**Table 5.** Project vs. process.

| Factor | Conflict Type /Field of Literature | Mechanism | Manifestation (of the Conflict Type) |
|---|---|---|---|
| Project vs. process | *Systems thinking* 'Will it work?' This is about coordination | Reactive and technical (project) rather than proactive and visionary (process) | No 'real boss' in the project |
| | Network coordination and cooperation | (No project lead meant) no coordinated effort to identify, let alone deal with uncertainty | Forced match |
| | Level of coordination, weak/strong | | Throwing the problem over the fence |

To illustrate, project management is about following the plan from start to end, like a cake recipe to put it crudely, while 'process' is about adapting and evolving to developments along the way. The project perspective 'drives out' the possibility of looking at uncertainties within organizations and restrains the development of exploring uncertainties

and strengthening of the collaboration between organizations in search of new knowledge to deal with uncertainties. The project drive reinforces the culture of the project and risks and drives out uncertainties as an attention point.

The field of literature linked to this concept is systems thinking, which provides insight into how joint infrastructure projects are large and complex. The collaboration and ILT project could be identified as the overall system, which consists of a combination of many subsystems (stakeholders and projects). One can think of one infrastructure operator in the project operating and developing its own project and working as its own system with their own rules, procedures, and experience. Together with many other operators, such as in the ILT, all these 'systems' (and subprojects) come together and bring their own idea on how they want to make their infrastructure project work. Collaborating on a joint project composed of a network of operators and other stakeholders and aligning all ideas and visions was a challenge.

The ILT was carried out through project management, which means that how the project team dealt with uncertainties (unforeseen events) was reactive and technical rather than proactive and engaging. In other words, if something happened, the stakeholders would react to events as they happened rather than trying to identify uncertainties beforehand, and focusing on figuring out how they would approach a joint problem together before it potentially happened. This is not the same as trying to plan for everything that could go wrong, because that is what risk management focuses on, but rather, they would agree upon the contours of a process on how to deal with a potential unforeseen event.

There was no 'real' project lead in the ILT. The municipality of Amsterdam initiated the project but did not exert any 'real' power over the stakeholders and could thus not take the lead. There was a horizontal hierarchy between the stakeholders, meaning that the organizations were practically equal. The ILT was also a forced match of stakeholders who had no choice but to participate, 'now or never', if they wanted their infrastructures in the Zuid-As district. They could not choose their partners. The difficulty of collaborating can be illustrated by how 'forced marriages are rarely successful'. Rather than jointly discussing challenges and exploring opportunities and possibilities, the issues they encountered were 'thrown over the fence'. Instead of opening a dialogue on how to approach an issue, the approach was instead to tell the infrastructure operator, or level of management, who was thought to bear responsibility to solve the issue, to 'fix it'. This in turn caused conflict and frustration with personnel of infrastructure operators being treated like this, and this happened both internally and externally between the operators.

To illustrate the thought process during the case study and the subsequent analysis, it is helpful to have an overview. Figure 1 shows a conceptual map with the connections of the core observations that were made during the research.

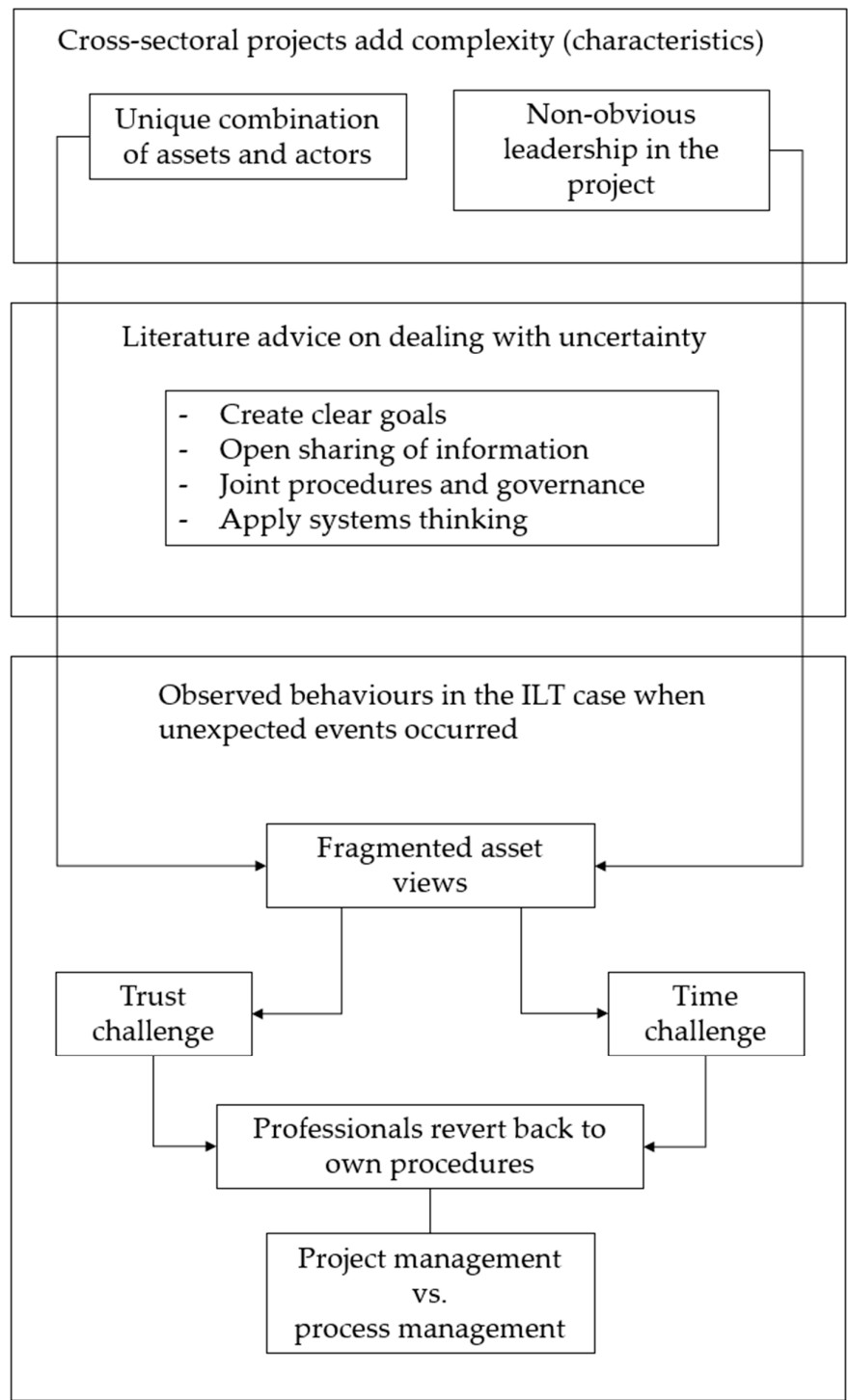

**Figure 1.** Conceptual map.

## 5. Discussion

Interestingly, when observing the tables from the analysis, the *types of conflict* are the most interesting takeaways. That does not mean the challenges of collaboration in the ILT would have been fixed if only systems thinking would have been applied and likewise information sharing, goal setting and joint procedures. It is much more difficult than that. What was observed is that these four conflicts were absent, or poorly represented, as well

as difficult to realize. When large infrastructure operators come together to work on a joint project, they bring with them their own perspectives and ways of thinking about systems, of how they share information, set goals (for themselves), and use well-established procedures that works for them. Bunch all the different perspectives from all the stakeholders together in a cross-sectoral project, and one can see that aligning all the perspectives is far harder to realize than the literature conveys. Creating shared goals and realizing trust is not easily established if procedures prove conflicting and project collaborators tell you what they expect you to do rather than engage in an open process toward understanding each other's uncertainties and invest in jointly seeking for solutions. There is not much in the literature at this moment that explains this paradoxical dichotomy, because the literature lacks empirical data on collaboration of cross-sectoral infrastructure projects and the governance of uncertainty.

The infrastructure operators had a mindset of requirement rather than engagement during their collaboration, and there seemed to be a focus on conceptual relationships rather than human relationships. However, in the case of ILT, conceptual relationships did not allow for the level of connection required for the participants to feel comfortable enough to open up and enter an 'actual' collaborative state where one can jointly take on interdependent challenges in the project. Consequently, none of the stakeholders could understand or grasp the whole 'picture' of the tunnel project from the start. Therefore, ILT showed how one cannot rely on something like 'requirement capturing' at the start of a project. Said differently, what the stakeholders in the ILT did was 'requirement posing' instead of eliciting and connecting the different stakeholders' contributions to the project. Instead of engagement together, the ILT project turned into a situation of each participant posing their demands without exploring the interdependencies toward the rest of the infrastructure assets of the other infrastructure operators. There was no integrated interpretation of the whole system of infrastructures inside the tunnel.

Even if one could make such a blueprint, there are power differences in a project. Some operators have a greater effect on the project than others, even in a project with a horizontal hierarchy. One example of exerting power is that one of the infrastructure operators was not satisfied with the municipality's design of the tunnel. The operator meant that the design did not consider the potential safety issues of their infrastructure asset(s) and demanded a new design that better suited their needs. The result of this challenge was that the infrastructure operator was given roughly eight months extra time to come up with a solution. This change had an impact on, e.g., the future process of connecting surrounding buildings to the infrastructures [51]. The power differences thus contribute further to causing project participants to gain an incomplete picture of the project as a whole, since certain demands overburden other types of demands. This can even result in demands falling between the cracks and facilitate the realization of an incomplete vision of the final result.

There was no project lead with any real power in the ILT. The municipality tried to put the right people together and somewhat coordinate them but could not do much more than that. There was no over-arching project management, or process management, no hierarchy, and no real mutual coordination. Project management literature would say something along the lines of 'put in a project, with project management, clearly assign tasks and responsibilities and the problem is solved.' If there is a hierarchy that coordinates, where someone is a project manager overlooking the functioning of the tunnel, with all its facilities and services, as well as having contracts with everyone working in there, then someone can coordinate the project to make it work as long as the individuals have the mechanisms to exert power over the stakeholders. However, in the ILT, there was no one with this power to take on the hierarchy. Knowing this full well, the municipality took on the financial burden of the project, covering expenses that exceeded 'normal' costs for the infrastructure operators. However, the municipality did not take on the project lead role.

Employees representing their infrastructure organization bring their own risk perceptions to the table and think that 'money is the problem', but when the municipality says, 'money is not the problem', the employees in infrastructure operators still think that 'money is the problem'. This is what happened in the ILT. People are engrained in their way of thinking and what mindset to have when they enter a project. So, when the municipality says that money is not a problem, it is very difficult to have people change their mindsets about money. Said differently, if a value problem is brought to the table, and everyone in the cross-collaboration has a different set of values at that table, it is near impossible to have people adopt a new mindset about money in such a project. Their excess focus on the finances also meant that they invested less time in technical solutions and interdependencies than they could have. In hindsight, some of the respondents regretted that they did not spend more time on the assets instead of on the financial aspect. When the infrastructure operators faced challenges related to the collaboration, they would rather talk about the issues that are high on their agenda, which arise from their method of working, rather than delving into the underlying issues of why they are not able to collaborate. This way they are not really fixing anything or are at best 'putting out fires' when they occur, leaving the uncomfortable part out. There is ambition to collaborate, sure, but there is something else going on 'underneath the surface' that is not addressed.

The design of the tunnel was already made by the municipality before they included the other stakeholders and 'forced' them to develop this solution. The municipality seems to have underestimated the complexity of the ILT project, because they did not fully know how the project would unfold. They had a concrete tunnel to seemingly make it easier to manage infrastructures, especially considering the future plans of the Zuid-As. Perrow [11] argues that propensity for 'normal' failure is partly hidden in the interactive complexity of technology and the organizations that are in control of them. A second dimension, according to Perrow [11], which influences the propensity for 'normal' failure is the level of coupling between the organizational and technical technological subsystems. Our analysis shows that even in seemingly one of the easiest forms of straightforward technological design such as a 'concrete tunnel with a few lines and pipes' in it and seemingly decoupled activities are in actual reality much more complex and intricately coupled. Thus, even seemingly straightforward infrastructure projects become sufficiently complex and coupled to defeat expectations of laymen and project routines employed by infrastructure organizations and their experts. ILT turned out to be a much harder project to execute than initially assumed. There was no process in place to identify interdependencies between the stakeholders' assets from the beginning of the project, and nobody cared or pushed the project collaboration to explore uncertainties resulting from them. The interdependencies were ignored, and existing ways of working were incapable of picking up signals of potential consequences of interdependencies for other operators, which is very hard in cross-sectoral projects since standard (project) practice is that everyone focuses on their own assets. A key value in project management is the assignment of clear responsibilities which in principle should not overlap. The interdependencies were not understood, and that is a key element of where uncertainties come from. Everyone stuck to their own solutions in terms of their process and their requirements, but no one communicated about the level of interconnectedness. The experts from the infrastructure operators only presented their contributions on the project management table; no one connected to investigate uncertainties resulting from interdependencies. There was no process in place to identify interdependencies between the stakeholders' assets from the beginning of the project nor did anyone make a joint design of the ILT.

The case showed how hard it was for those involved to break out of their own perspective. The most prominent element was that even though the municipality regularly claimed to take the financial risk, several stakeholders reverted to perceived financial risks as reason for objection to certain solutions. This was an example of how a *value prevalence* (costs) of a stakeholder is hard to change in a cross-sectoral context. In addition, the

interdependencies between the subsystems brought in by the different stakeholders were orphaned: for a long time, no one could oversee or felt responsible to oversee the interactions between the infrastructures. This was an example of how a *system focus* is hard to change in a system-of-system [52,53] context.

In addition, one could argue that the literature on (cross-)sectoral innovations does not focus on actual operations including the analyses of the governance of uncertainties. Instead, this body of literature usually recites how a project was realized from the project perspective, rather than how infrastructures projects were realized from an infrastructure operator's perspective, nor how the process of bringing together perspectives from different stakeholders in the exploration phase of the project actually unfolded.

The findings of this research are context specific, and several criticisms can be forwarded. The limited number of people available for interviews, possible hindsight bias of the interviewees, and the age of the project were limiting factors. There were also information gaps in the case data, and the data did not evenly cover all the participating organizations. In other words, some organizations were able to provide more insight to this research than other organizations. Despite these limitations, it was still important to study the ILT case, and choosing an older case had its merits. The case was de-politicized, and consequently, the interviewees were more comfortable with sharing their experiences and opinions on the collaboration. It was also revealed that the organizations do not show much interest in looking back to learn from each project. Before the infrastructure operators are finished with one project, they have already moved on to the next one. There is in other words an overlap of projects, and a lack of project evaluation of finished projects, also in the scientific literature [54,55].

We have seen how literature, in the context of process management, proposes more generic ways in which, in early stages of decision making, the stakeholders create a more shared perspective on the asset, on the value prevalence, on the way decisions will have to be made. In an engineering context, we would suggest the term 'empathic engineering' and define the concept as a form of engineering decision making in the early stages of a project in which the focus is on open and connected dialogue between the stakeholders. What we see in the ILT case is that the stakeholders tend to bring requirements without the inclination to develop an asset-wide understanding of the effect of those requirements, their requirements on others, or other requirements on the elements of the asset that they feel a connection to. See Table 6 for an initial comparison between traditional systems engineering and empathic engineering approaches.

**Table 6.** Comparison of systems engineering and empathic engineering.

| Systems Engineering | Empathic Engineering |
| --- | --- |
| Validation of in decomposed (sub)system choices | Discovery of decomposed (sub)system interactions |
| System-of-system contexts with understood interfaces | System-of-system contexts with uncommon interfaces |
| Integration of systems through vertical coordination | Integration of systems through horizontal coordination |
| Engineering integration as separate activity | Engineering integration as shared activity |
| Knowledge sharing driven | Knowledge discovery driven |
| Project manager in the lead, stakeholders follow | Project manager facilitates, stakeholders lead |

In more 'normal', repetitive, and sectoral projects, the integration of different elements can be completed as a separate engineering activity, as most of the interactions between system (or subsystems) are understood. However, in cross-sectoral projects, with their unique character, the integration of engineered elements should be seen far more as a shared engineering activity. This also has a consequence for the way that this process is managed: this requires far more a process managerial style in which the different stakeholders are seen as contributing equals than a project managerial style in which different stakeholders could be seen as to be controlled entities. It also requires different thinking from those stakeholders and the need to invest in a trusted, curious, and open engineering

phase, which we here will call empathic engineering. This is a phase where everyone involved is probably underestimating the complexity of the overall asset, underestimating the contribution to that complexity by their technology and ways of working for other subsystems.

## 6. Conclusions

During the investigation on how infrastructure operators cope with uncertainty in the joint project of the ILT, four conflicts were identified. These conflicts highlight the added complexity of cross-infrastructural projects. When comparing the results with the literature, it was found that these conflicts in this setting are not very well researched yet. It is not reported much on, both in the literature and among infrastructure professionals, and as a result, it is ill understood. What the literature offers in terms of ways of working or key concepts to explain ongoing work is also not immediately applicable to a complex cross-infrastructural project, because collaboration is much more complex in a real setting such as the ILT. Obviously, the study of the ILT is a single case study and thus cannot be generalized to fit all cases of a similar nature. The case study is explorative but raises some interesting issues with a focus on the conceptual findings rather than the empirical data. The study merely illustrates the characteristics of cross-sectoral complexity. It highlights the specific uncertainties and the consequences in decision making that follow from that. Our proposition would be that other cross-infrastructural projects will reveal similar results. It is not expected that the findings of this article can be replicated in other projects, but it is expected that the conceptual findings can. This requires much more attention in research to understand how people from completely different sectors, with entirely different backgrounds, in context without a clear hierarchical relation must make decisions in those situations when infrastructures connect in other than standardized ways. Existing forms of engineering decision making, existing project management, and existing governance of uncertainty seem to be ill prepared for the dynamic complexity that follows from interdependencies in infrastructure projects going through major transitions: this is a research challenge that requires attention. The proposed concept of empathic engineering is a part of a possible solution on how to govern uncertainty in project settings such as, e.g., the ILT. Because different types of cases will reveal different types of challenges on how to deal with uncertainty, it may certainly be interesting to follow up on the impact on infrastructure interdependency in future research.

One of the most significant findings of this research is that infrastructure operators enter a cross-sectoral project but end up working in isolation on their own subproject, but even more important is the consequence of this isolation; uncertainties are not discussed or negotiated. The ILT was a joint project, so it was assumed from a research point of view that challenges affecting multiple infrastructures would be proactively scanned and dealt with across organizational boundaries, but the ILT was indeed without 'real' collaboration. It was not surprising to see that the organizations worked in isolation. The surprising part was that this is a well-known challenge for the infrastructure operators, as well as in the literature, and it seems to be a dominant discussion point for them in hindsight. Despite knowing this, they still did not work toward a solution to this issue.

**Author Contributions:** Conceptualization, O.P.R., M.L.C.d.B., and W.W.V.; methodology, O.P.R., M.L.C.d.B., and W.W.V.; investigation, O.P.R.; writing—original draft preparation, O.P.R.; writing—review and editing, M.L.C.d.B., and W.W.V.; supervision, M.L.C.d.B., and W.W.V.; funding acquisition, M.L.C.d.B., and W.W.V.. All authors have read and agreed to the published version of the manuscript.

**Funding:** This research was funded by Nederlandse Organisatie voor Wetenschappelijk Onderzoek, the Dutch National Science Foundation in the Responsive Innovations Program (with project number [439.16.823]), co-funded by NGinfra.

**Institutional Review Board Statement:** Not applicable.

**Informed Consent Statement:** Informed consent was obtained from all subjects involved in the study. Ethical approval was granted, with approval code 1447.

**Data Availability Statement:** The data presented in this study are available on request from the corresponding author. The data are not publicly available because personal information collected which can identify the interviewee will not be shared beyond the study team. Personal information and quotes can only be used with the written consent of the interviewee.

**Acknowledgments:** We want to thank all interviewees who shared their views openly on the ILT project.

**Conflicts of Interest:** The authors declare no conflict of interest.

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
