# Peer review of "Dealing with Cross-Sectoral Uncertainty: A Case Study on Governing Uncertainty for Infrastructures in Transition"

_sustainability, doi:10.3390/su15043750_

Round 1

Reviewer 1 Report

The paper reflects on the complexity of cross-sectoral projects using a specific case study in de Zuid-As, Amsterdam, the Netherlands. The paper mentions the concept of "emphatic engineering" although is not well-defined in the body of the paper. Here are specific comments to be addressed by the authors:

1. Define "Empathic Engineering" in the introduction and provide background information supported by references.

2. Define "Uncertainty" from the paper's perspective. Provide background information supported by references.

3. The literature review focused on goal setting, information sharing, joint procedures, systems thinking. What is the relationship with "Empathic Engineering" and "Uncertainty" ? Please expand the literature review. A conceptual diagram may be useful to understand the relationship among them.

4. Tables presented in section "4. Results" are not specifically described in the narrative. Please review and explain the purpose and contributions, and findings from each table.

5. How are the factors presented in tables 2 through 5 can be quantified in a decision-making process ? Please be more specific on what kind of decisions/outcome are affected by these factors by providing examples.

6. Systems thinking, Information sharing, goal setting, and joint procedures are mentioned in the Discussion section as part of the solution but insufficient due the complexity of the projects. The paper talk about "different perspectives from all the stakeholders", but the authors do not specify how to specific address it and integrate it into the decision making process.  A flowchart may help to better described the process.

7. In the Conclusions, the authors mention that : "The most significant finding of this research is that infrastructure operators enter a cross-sectoral project but end up working in isolation on their own sub-project". This finding is not new.  Be more specific, how the authors considered the problem will be solved.

8. Describe the limitations of the current research and provide ideas for future work.

9. Based on the previous comments, the title of the paper can be misleading and may be better include the name of the specific case study in the title.

Reviewer 2 Report

The authors present a well-structured paper on the problems of uncertainty in the implementation of a cross-sector infrastructure project: the integrated cable and pipeline tunnel (ILT) in the Zuid-As area in Amsterdam, The Netherlands. For this purpose, a single research question was formulated in the introductory section.

However, for the acceptance of the work, some major and minor changes should be undertaken for the authors. Some of them are as follows:

Title of the work

The topic of the work should be reformulated, it concerns a specific case, which should be emphasized in the title.

Abstract

The Abstract does not actually present any findings of the work that could stimulate interest in reading it. I recommend including specific conclusions or at least interest in the research topic undertaken.

One of the biggest shortcomings of the work is that it refers to only one case study. As a result, it is not a universal study, so I am not sure whether it is justified to put the conclusions formulated by the authors. What is the authors' other experience with the issues addressed in the paper?

Discussion

In my opinion discussion should also focus on a critical evaluation of one's own research results. In conclusion, there is a lack of critical assessment of the results achieved. 

Reviewer 3 Report

1. IN LINES 99 AND 102 BETTER TO MENTION THE AUTHOR'S NAME BEFORE REFERENCE NO. 11

2. IN CONCLUSION RECOMMEND TO SUGGEST WAYS FOR IMPROVING THE COLLABORATION OF A JOINT INFRASTRUCTURAL PROJECT

Round 2

Reviewer 1 Report

Most of the previous comments have been addressed in the revised version.

6. Systems thinking, Information sharing, goal setting, and joint procedures are mentioned in the Discussion section as part of the solution but insufficient due the complexity of the projects. The paper talk about "different perspectives from all the stakeholders", but the authors do not specify
how to specific address it and integrate it into the decision making process. A flowchart may help to better described the process.

RESPONSE
This is a much-appreciated suggestion. To be fair, as a part of this explorative type of research, a flowchart is already a few steps ahead of where we are in our research. The topics of systems thinking, information sharing, goal setting, and joint procedures are the literature fields that we
have connected to the identified factors of our analysis and are not considered solutions. They are, however, literature fields in which countless types of solutions can be found. We think that the type of solution that will be beneficial will differ from project to project, as they are context specific. The case study of this article does not give a general layout of a solution, but rather an idea that the solution(s) can be found in the four types of literatures. We have an interpretative and explorative approach where we used the case study as a new way of looking into the complexity of how organizations collaborate regarding the governance of uncertainties.

COMMENT TO ADDRESS: Based on the response, a CONCEPT MAP or a MIND MAP will be able to illustrate the relationships among the topics discussed in the paper. It would be able to consolidate/summarize all the explanations in the narrative.

There is a need to clarify the relationships of the concepts described in the paper. A concept map or a mind map is recommended for this purpose.
